# Antiviral Agents as Therapeutic Strategies Against Cytomegalovirus Infections

**DOI:** 10.3390/v12010021

**Published:** 2019-12-23

**Authors:** Shiu-Jau Chen, Shao-Cheng Wang, Yuan-Chuan Chen

**Affiliations:** 1Department of Neurosurgery, Mackay Memorial Hospital, Taipei 10491, Taiwan; chenshiujau@gmail.com; 2Department of Medicine, Mackay Medicine College, Taipei 25245, Taiwan; 3Jianan Psychiatric Center, Ministry of Health and Welfare, Tainan 71742, Taiwan; WShaocheng@gmail.com; 4Department of Mental Health, Johns Hopkins Bloomberg School of Public Health, Baltimore, MD 21205, USA; 5Program in Comparative Biochemistry, University of California, Berkeley, CA 94720, USA

**Keywords:** cytomegalovirus, acute/latent infection, congenital infection, antiviral agent, therapeutic strategies, nucleic acid-based therapeutic approach, HCMV vaccine, adoptive cell therapy

## Abstract

Cytomegalovirus (CMV) is a threat to human health in the world, particularly for immunologically weak patients. CMV may cause opportunistic infections, congenital infections and central nervous system infections. CMV infections are difficult to treat due to their specific life cycles, mutation, and latency characteristic. Despite recent advances, current drugs used for treating active CMV infections are limited in their efficacy, and the eradication of latent infections is impossible. Current antiviral agents which target the UL54 DNA polymerase are restricted because of nephrotoxicity and viral resistance. CMV also cannot be prevented or eliminated with a vaccine. Fortunately, letermovir which targets the human CMV (HCMV) terminase complex has been recently approved to treat CMV infections in humans. The growing point is developing antiviral agents against both lytically and latently infected cells. The nucleic acid-based therapeutic approaches including the external guide sequences (EGSs)-RNase, the clustered regularly interspaced short palindromic repeats (CRISPR)/CRISPR-associated protein 9 (Cas9) system and transcription activator-like effector nucleases (TALENs) are being explored to remove acute and/or latent CMV infections. HCMV vaccine is being developed for prophylaxis. Additionally, adoptive T cell therapy (ACT) has been experimentally used to combate drug-resistant and recurrent CMV in patients after cell and/or organ transplantation. Developing antiviral agents is promising in this area to obtain fruitful outcomes and to have a great impact on humans for the therapy of CMV infections.

## 1. Introduction

### 1.1. Cytomegalovirus Overview

Cytomegalovirus (CMV) is a genus of *Herpesvirus* in the order Herpesvirales, in the family Herpesviridae, and in the subfamily Betaherpesvirinae. There are nine distinct human herpesvirus (HHV) species known to cause human diseases such as HHV-1, HHV-2, HHV-3, HHV-4, HHV-5, HHV-6A, HHV-6B, HHV-7, HHV-8 [1,2]. Human cytomegalovirus (HCMV, HHV-5), with a double-stranded DNA genome of about 230 kb, is the most studied one among all CMV. HCMV usually causes moderate or subclinical diseases in immunocompetent adults; however, it may lead to opportunistic infections to affect individuals whose immune functions are compromised or immature [3,4]. The primary target cells of HCMV are monocytes, lymphocytes, and epithelial cells, and its major sites of latency are peripheral monocytes and CD34+ progenitor cells. HCMV infection causes a broad range of diseases such as pneumonia, retinitis, gastrointestinal diseases, mental retardation and vascular disorders, and is a major cause of morbidity and mortality for humans [5,6,7]. After infection, HCMV is recurrent and competent to remain latent within the body over long periods [5,6]. In all patients, the reactivation of latent HCMV can damage tissues and lead to organ disease, and reactivated CMV may trigger indirect immunomodulatory effects to cause detrimental outcomes, including increased mortality and graft rejection of organ transplantation in recipients [8]. Furthermore, congenital infection is a major problem with HCMV in that it can result in a severe cytomegalic inclusion disease of the neonate, mucoepidermoid carcinoma, and other malignancies eventually [7,9].

### 1.2. CMV Molecular Biology

CMV structure mainly consist of DNA core, capsid, tegument and envelope from inside to outside. The genome is complexed helically to form a DNA core, which is enclosed in a capsid composed of a total of 162 capsomere protein subunits. The capsid with a diameter of 100 nm is surrounded by the tegument. The tegument is enclosed by a lipid bilayer envelope containing viral glycoproteins to give a final diameter of about 180 nm for mature infectious viral particles (virions) [10]. The tegument compartment contains most of the viral proteins, with the most abundant one being the lower matrix phosphoprotein 65 (pp65) which is also referred as unique long 83 (UL83). The function of the tegument proteins can be classified as follows: (1) proteins that play an important role for the assembly of virions during proliferation and the disassembly of the virions during entry (structural use) and (2) proteins that modulate the host cell responses for viral infection (non-structural use) [11]. The viral envelope surrounding the tegument contains more than 20 glycoproteins that are involved in the attachment and penetration of host cells. These structural proteins include glycoprotein B, H, L, M, N, and O. CMV productive infection results in the coordinated synthesis of proteins in three overlapping phases according to the time of synthesis after infection, that is, immediate-early (0 to 2 h), early (<24 h), and late (>24 h) viral proteins which are expressed by immediate-early, early and late genes, respectively [11]. Immediately after CMV infections, the immediate early genes transcribe and ensure the transcription of early genes, which encode proteins required for the viral replication. The late genes mainly code for structural proteins.

### 1.3. CMV Life Cycle

CMV infection will start once a virion attaches a host cell with specific receptors on the cellular surface. For a lytic infection pathway, following linking of viral envelope glycoproteins to host cell membrane receptors, the virions enter the hosts by receptor-mediated endocytosis and membrane fusion. The viral capsid decomposes to release viral DNA genome to manipulates host enzyme systems to make new virions. During symptomatic infection, infected cells express lytic genes to demonstrate a lytic pathway [12]. Nevertheless, instead of this, some viral genes may transcribe latency associated transcripts to accumulate in host cells. In this pattern, viruses can persist in host cells indefinitely to have a latent infection pathway. The primary infection may be accompanied by limited illness and long-term latency is often asymptomatic. For the lysogenic pathway, the viruses are persistent in the host, not causing any adverse reactions, but can be transmitted to other hosts by direct contact. When CMV are stimulated by explanation or their host immune system is suppressed, the dormant viruses can reactivate to begin generating large number of viral progenies to cause symptoms and diseases, described as the lytic life cycle [12,13].

Viral latency can be divided into two models, namely proviral latency and episomal latency. CMV is defined as the episomal latency model which is essentially quiescent in myeloid progenitor cells. It can be reactivated by differentiation, inflammation, immunosuppression or critical diseases [14]. Latency is a specific phase in CMV life cycles in which virions stop producing posterior to infection, but the viral genome has not been entirely removed from host cells, that is, CMV latency is referred to as the absence of virions, despite the detection of viral DNA in hosts. In some clinical cases, the reactivation of latent infections is likely to lead to health risk. The molecular mechanisms by which latency is established and maintained have been explored. However, our understanding of the biology of CMV latency and reactivation at the molecular level would be significantly strengthened through analyses of both experimental and natural latency using systematic approaches [14]. 

## 2. CMV Infection

### 2.1. Signs, Symptoms and Complications

Most CMV infections are silent and CMV rarely causes signs or symptoms in healthy people. Though CMV infection is usually ignored in healthy people, the diseases can be life-threatening for the immunocompromised, immunosuppressed and immunonaive patients, such as newborn infants, the elderly, the sick, acquired immunodeficiency syndromes (AIDS) patients, and organ transplant recipients. A mother who acquires an acute CMV infection during pregnancy can transmit viruses to her baby, and thereby the baby might experience signs and symptoms. People at higher risk of CMV infections encompass newborns infected through their mothers before birth, babies infected through breast milk and people with weakened immune systems such as organ transplantation recipients or immunodeficient patients. The major signs, symptoms and complications which CMV influence all individuals including healthy adults, people with weakened immunity and babies are shown in Table 1.

### 2.2. Congenital Infection and Sequelae

CMV is transmitted by close interpersonal contact such as saliva, semen, urine, breast milk, or vertically transmission which viruses pass the placenta and directly infect the fetus [15,16]. CMV is the leading cause of congenital viral infection [17,18,19,20]. CMV infection is mostly or mildly asymptomatic among the general population (85%–90%). However, around 10%–15% of infants with the congenital infection may be at risk of sequelae such as mental retardation, jaundice, hepatosplenomegaly, microcephaly, hearing impairment and thrombocytopenia [21,22,23,24]. Among the above sequelae, the most devastating one is the central nervous system (CNS) sequelae related to neurodevelopment in that CNS injury is irreversible and persists for life, including mental retardation, seizures, hearing loss, ocular abnormalities and cognitive impairment [25,26,27]. That means the asymptomatic newborns with CMV infection still have an increased risk for long-term sequelaes, especially, mental retardation and sensorineural hearing loss (SNHL) [28,29,30,31], making CMV the leading nonhereditary cause of SNHL [24,32]. CMV can undermine both adaptive and innate immunity, silencing natural killer (NK) cells and inhibiting T cells to present viral antigens [33,34,35]. 

## 3. CMV Anti-Viral Drugs

At present, some antiviral drugs have been approved for the treatment of CMV infections clinically. Current available drugs for antiviral therapy of CMV infections include the inhibitors of viral DNA polymerase, such as the nucleoside analog ganciclovir, the nucleotide analog cidofovir, and the pyrophosphate analogue foscarnet [36]. All these drugs have low oral bioavailability and dose-related toxicities, and therefore new antiviral agents with improved efficacy and fewer side effects need to be developed. Several drugs with anti-HCMV activity are preclinically or clinically evaluated, including a series of benzimidazole riboside compounds showing efficient inhibition in the process of HCMV replication such as genomic DNA maturation. Another attractive inhibitor candidate was the phosphorothioate oligonucleotide fomivirsen, which specifically binds to sequences complementary to CMV major immediate-early transcription sites so that it inhibits the viral gene expression. However, these inhibitor compounds are currently waiting for further examination before they can be used in clinics [17]. 

Currently, ganciclovir is still the first treatment of choice for CMV infections. Letermovir has been approved for the prophylaxis of CMV infections in patients. Several new drugs were developed but still failed in the phase III and more clinical trials would be needed, including maribavir and brincidofovir [37]. Valnoctamide, a neuroactive mood stabilizer which inhibits CMV infection in the developing brain and attenuates neurobehavioral dysfunctions, was shown to have anti-CMV potential [38].

### 3.1. Letermovir

Letermovir is a novel antiviral drug which has been approved by the USA Food and Drug Administration (FDA) through a fast track procedure and granted as an orphan drug by the European Medicines Agency (EMA). The drug was tested in CMV infected patients and likely be useful for other patients who had organ transplantation or human immunodeficient virus (HIV) infections [39]. It has been clinically applied for CMV prophylaxis or treatment in hematopoietic stem cell recipients, thoracic organ recipients and lung transplantation recipients [40,41,42]. Letermovir has several advantages over conventional CMV antiviral agents as follows. Firstly, it can be given orally, so hospitalization and intravenous injection are not needed. Secondly, it is mild in toxicity, not related to myelotoxicity and nephrotoxicity [43,44]. Thirdly, it targets the CMV terminase complex instead of CMV DNA polymerase, so there is no risk to induce cross-resistance with existing anti-CMV drugs [44]. However, the CMV antiviral therapy will finally fail and acquired antiviral drug resistance is not avoidable if there is no immune control [45]. It should be noted that more data are required to provide insights of the mutations detected in vivo, interpretation of genotyping results, and outcomes of the clinical correlation. To provide useful information, it would be recommended to establish databases for the surveillance and interpretation of resistance for CMV [36]. 

### 3.2. Maribavir

Maribavir is a promising anti-HCMV compound which is administered orally; however, it is still under advanced clinical trials. The drug targets the viral kinase UL97 which is crucial for the formation of viral teguments and assembly complexes for virion releasing [36]. However, it is not recommended to co-administer maribavir and ganciclovir both in that maribavir is an inhibitor of the UL97 enzyme which is required for the assimilation of ganciclovir. Maribavir potentially substitutes for other traditional anti-HCMV drugs because of its reduced haematotoxicity and nephrotoxicity compared with ganciclovir and valganciclovir [36].

Maertens et al. used maribavir (dose-blinded) to treat cytomegalovirus reactivation preemptively for recipients of hematopoietic cell or solid organ transplants (SOT) (≥18 years old) with CMV reactivation in a phase II and open-label clinical trial [46]. The results showed that maribavir at a dose of at least 400 mg twice daily had efficacy like that of valganciclovir for removing CMV viremia. Though a higher incidence of gastrointestinal adverse events were found in the maribavir -treated group, the neutropenia incidence was lower [46]. 

Papanicolaou et al. used dose-blinded maribavir 400, 800, or 1200 mg twice-daily for up to 24 weeks to treat hematopoietic-cell or SOT recipients (≥12 years old) with refractory or resistant CMV infections in a phase II and double-blind clinical trial [47]. The result revealed that it was active to against refractory or resistant CMV infections using maribavir more than 400 mg twice daily in transplant recipients and no new safety signals were identified in this trial [47].

## 4. CMV Inhibition by Nucleic Acid-Based Therapeutic Approaches

The treatment of diseases caused by CMV is quite challenging because of high mutation rates and latency. Thus, infection is still a serious threat to humans. Fortunately, external guide sequences (EGSs), transcription activator-like effectors nucleases (TALENs) and the clustered regularly interspaced short palindromic repeats (CRISPRs)/CRISPR-associated 9 (Cas9) nuclease system might provide effective therapeutic strategies to treat diseases caused by CMV through designing a specific DNA or RNA sequence that target essential genes for viral growth. However, the effective modification of the viral genome avoiding off-target effects and the option of escape variants ignoring the editing of these approaches are required for successful clinical application.

### 4.1. EGS-RNase

Ribonuclease P (RNase P) is a unique RNases in that it is a ribozyme – an RNA that acts as a catalyst which is somewhat like a protein enzyme. Its function is to cut an extra or precursor sequence of RNA on transfer RNA (tRNA) molecules; that is, to catalyze the cleavage of precursor tRNA into active tRNA without any protein component. RNase P has the activity in cleaving the 5’ leader sequence of precursor tRNA. EGSs signify the short RNAs that induce RNase P to specifically cleave a target mRNA by forming a precursor tRNA-like complex. Therefore, EGS technology probably acts as an effective strategy for gene-targeting therapy. 

Deng et al. reported that engineered EGS variants induced RNase P to efficiently hydrolyze target mRNAs which code for HCMV major capsid protein [48]. In vitro, the engineered EGS variant was more efficient in inducing human RNase P-mediated cleavage of the target mRNA than a natural tRNA-derived EGS by about 80-fold. In cells infected with HCMV, the EGS variant and natural EGSs resulted in HCMV gene expression reduction rate by about 98% and 73%, and the viral growth was inhibited by about 10,000 and 200-fold, respectively. The results showed that the EGS variant has higher efficiency in blocking the expression of HCMV genes and viral growth, compared with the natural EGS [48].

Li et al. explored the antiviral effects of an engineered EGS variant in targeting the shared mRNA sequence which codes for capsid scaffolding proteins (mCSP) and assemblins of murine CMV (MCMV) in the animals [49]. In vitro, the EGS variant was more active in directing RNase P cleavage of the target mRNA than a natural tRNA by 60-fold. In MCMV-infected cells, the EGS variant decreased mCSP expression by about 92% and inhibited viral growth by about 8000-fold. In MCMV-infected mice, the EGS variants were more effective in reducing mCSP expression, decreasing viral production, and increasing animal survival, compared with the natural EGS. The results demonstrated that the EGS variant with higher targeting activity in vitro are also more effective in inhibiting MCMV gene expression in mice [49].

### 4.2. CRISPR/Cas9

In CRISPR/Cas9 system, CRISPR is used to build RNA-guided genes drives to target a specific DNA sequence. By the Cas proteins and a specifically designed single-guiding RNA (sgRNA), the genome can be cut at most locations with only the limitation of a protospacer adjacent motif (PAM) sequence (NGG) existing in the target site [12]. CRISPR/Cas9 has been extensively used as an effective technique of gene editing for engineering or modifying specific genes. It was shown to successfully work as an efficient genome editing tool in a wide range of organisms including HCMV [50]. Consequently, it hints that the CRISPR/Cas9 can be a potential antiviral agent for the treatment of CMV infections.

Gergen et al. designed two CRISPR/Cas9 systems which contain three sgRNAs to target the HCMV UL122/123 gene crucial for the regulation of lytic replication and reactivation from latency [51]. Both systems caused mutations in the target gene and an accompanying reduction of immediate early gene expression in primary fibroblasts. The singleplex strategy caused 50% of insertions and/or deletions (indels) in the viral genome to appear in further detailed analyses in U-251 MG cells, resulting in a reduction in immediately early protein production. The multiplex strategy cleaved the immediate early gene in 90% of viral genomes and thereby inhibited immediate early gene expression. Therefore, viral genome replication and late protein expression were reduced by 90%. The multiplex CRISPR/Cas9 system can target the HCMV UL122/123 gene efficiently and prevent viral replication significantly [51].

van Diemen et al. observed that the clear depletion emerged in the anti-HCMV sgRNA expressing cells targeting essential genes UL57 and UL70 (1.3% and 4.6% mutants with frameshifts, respectively), compared with the sgRNAs targeting the nonessential genes US7 and US11 (83.5% and 85.8% mutants with frameshifts, respectively) [52]. The sequence complexity of the mutants selected upon UL57 and UL70 targeting was low, this suggested the selection of few suitable variants and subsequent expansion of infectious mutants need a lot of time. The results showed that the CRISPR/Cas9 is a promising strategy to restrict HCMV replication [52]. 

### 4.3. TALENs

Transcription activator-like effectors (TALEs) are crucial virulence factors that function as transcriptional activators in the cell nucleus of plants, where they directly bind to DNA via a central domain of tandem repeats [53]. Currently, TALENs were shown to be an effective tool for precise genome engineering with low toxicity and could be engineered to adapt for an antiviral strategy [54]. Hence, TALENs are likely to become part of a new approach for the treatment of CMV infections.

Chen et al. utilized three pairs of TALEN plasmids (MCMV1–2, 3–4, and 5–6) to target the MCMV M80 and M80.5 overlapping (M80/80.5) sequence to test their efficacy in blocking MCMV lytic replication in NIH3T3 cells [54]. Using lipofectamine or a specific lipoid NKS11 as transfection reagents, TALEN plasmids could specifically target the M80/80.5 sequence and effectively inhibit MCMV growth in cell culture when the plasmid transfection is prior to the MCMV infection. Using NKS11 which was previously proved to be nontoxic to mice as a transfection reagent, the most specific pairs of TALEN plasmids (MCMV3–4) showed that its competency to inhibit the replication and gene expression of latent MCMV in immunocompetent Balb/c mice. The administration of MCMV3–4 plasmids resulted in significant reduction in the copy number level of immediately early gene-1 DNA which is key to viral latency in mice, compared with the controls. Additionally, the innate immune DNA-sensing pathways of host might be involved in the induction of cytokine secretion such as type I interferon (IFN) (mainly IFN α and β) to fight against invading viruses. The result hinted that TALENs were able to provide an effective strategy to clear latent MCMV in animals [54].

## 5. HCMV Vaccines 

The vaccines against HCMV are still being developed, no licensed vaccine is available so far. It is necessary to have a specific and strong antibody and cell-mediated immunity to confer protection against HCMV primary infection through the analysis of the immune response to HCMV. Many efforts have been made to produce an HCMV vaccine for years, but a successful vaccine candidate has not yet to be developed, probably due to what immune responses needed for protecting against HCMV infections are still poorly understood [36]. To develop an effective HCMV vaccine, immune responses required to fight against HCMV and how to enhance these specific immune responses requires further study.

Choi et al. used the guinea pig which is a small-animal model to develop a CMV vaccine. A glycoprotein pentamer complex encoded by guinea pig cytomegalovirus (GPCMV) is essential for viral entry into non-fibroblast cells to enable congenital CMV [55]. Like HCMV, GPCMV needs a guinea pig specific cell receptor (platelet-derived growth factor receptor α) for fibroblast entry, but other receptors are required for non-fibroblast cells. A disabled infectious GPCMV vaccine strain induced humoral immune responses against viral pentamers to promote neutralization on non-fibroblast cells; thus, the vaccinated guinea pigs were protected from congenital CMV infections. The design including the pentamer complex as a part of vaccines may significantly enhance efficacy. This new finding lays stress on the importance of the immune response to the pentamer complex in contributing to the protection against congenital CMV and has opened a new era for the development of CMV vaccines [55].

Liu et al. hypothesized that a vaccine candidate able to elicit immune responses analogous to those of HCMV-seropositive subjects may confer protection against congenital HCMV [56]. The V160 vaccine has been shown to be safe and immunogenic in HCMV-seronegative humans, inducing both humoral and cell-mediated immune responses. In this study, they further demonstrated that sera from V160-immunized HCMV-seronegative subjects had similar quality attributes to those from seropositive subjects, including high avidity antibodies to viral antigens. This vaccine is a promising candidate against HCMV, but further evaluation in clinics for the prevention of congenital HCMV is required to warrant its safety, efficiency, and effectiveness [56]. 

## 6. Adoptive Cell Therapy for CMV Infections 

It is known that the infection-related morbidity and mortality will be increased, if the T cell-mediated immune responses are impaired in transplantation recipients. Virus-specific T cells capable of targeting a variety of pathogens in patients after hematopoietic stem cell transplantation (HSCT) have demonstrated potential efficacy for multiple viruses such as CMV, Epstein-Barr Virus (EBV) and adenovirus [57]. Adoptive T cell therapy (ACT), a type of immunotherapy in which T cells are given to a patient to treat diseases, has been developed to fight against drug resistant and recurrent CMV in SOT recipients. Therefore, ACT has become one of the therapeutic strategies for CMV reactivations in patients undergoing allogeneic HSCT and SOT. 

Faist et al. used the peptide specific proliferation assay (PSPA) to study CMV specific central memory T cells (TCM) repertoires and determined their functional and reproductive abilities in vitro [58]. In the animal model, the pathogen-specific TCM has demonstrated to have protective ability even at low numbers and could survive for long-term, proliferate extensively and show high plasticity after adoptive transfer. Though the clinical data showed that minimal doses of purified human CMV epitope-specific T cells are competent to remove viremia, it is still necessary to evaluate whether the human virus-specific TCM shows the same characteristic for ACT as mice. The results concluded that TCM had potential for prophylactic low-dose ACT. These good manufacturing practice (GMP)-compatible TCM could be used as a broad-spectrum antiviral T cell prophylaxis in allogeneic HSCT patients. In addition, PSPA would be a necessary tool for TCM characterization further during simultaneous immune monitoring [58].

Smith et al. applied high-throughput T cell receptor Vβ sequencing and T cell functional profiling to demonstrate the influence of ACT on T cell repertoire remodeling in the pretherapy immunity and ACT products [59]. The clinical response was consistent with significant changes in the T cell receptor Vβ landscape after therapy. This reconstitution was related to the emergence of effector memory T cells in responding patients, while nonresponding patients showed dramatic pretherapy T cell expansions with minimal change following ACT. The results revealed that immunological modulation following ACT required significant repertoire remodeling which might be damaged in nonresponding patients on account of the preexisting immune environment. Immunological interventions which controlled this environment were likely to improve clinical outcomes. ACT appears to be an advantageous strategy to restore immunological control against CMV affecting immunosuppressed patients such as SOT recipients [59].

## 7. Conclusions 

CMV is the most frequent etiological factor for congenital infections and its infections are still global health problems of humans. Though CMV infections are often opportunistic, they sometimes cause serious diseases in healthy adults with weakened immunity and babies. Currently, no drugs are available for asymptomatic infants and for infants with CMV congenital infections to reduce related morbidity during the neonatal period. Some traditional antiviral drugs (e.g., ganciclovir, valganciclovir, cidofovir, foscarnet) are applied for the treatment of CMV acute infections, however, their efficacy is limited by side effects, cross-resistance and others. There is no effective cure for CMV infections, especially for latent infections. Promisingly, many novel antiviral drugs/agents and preventive/therapeutic strategies have been approved for clinical application (e.g., letermovir) or are being developed (e.g., maribavir, EGS-RNase, CRISPR/Cas9, TALENs, HCMV vaccine, ACT). We should investigate the interactions between CMV and hosts thoroughly to understand how antiviral agents or therapeutic strategies affect CMV infection outcomes. Moreover, the development and implications of novel antiviral agents and preventive/therapeutic strategies should be explored as extensively as possible. The future research tendency and application of these new insights should also be a highlight and could potentially become a promising milestone in the development of therapeutic strategies for CMV infections.

## Figures and Tables

**Table 1 viruses-12-00021-t001:** CMV influence on individuals.

Individual	Major Signs and Symptoms	Complications
Healthy adult	Fatigue, fever, sore throat, muscle aches	problems with the digestive system, liver, brain and nervous system
People with weakened immunity	Problems affecting eyes, lungs, liver, esophagus, stomach, intestines, brain	Vision loss due to the retinitis inflammation, digestive system problems including inflammation of the colon, esophagus and liver, nervous system problems including encephalitis and myelitis, pneumonia
Baby	Premature birth, low birth weight, jaundice (yellow skin and eyes), enlarged and poor liver function, purple skin splotches and/or rashes, microencephaly (abnormally small head), enlarged spleen, pneumonia, seizures	Hearing loss, intellectual disability, vision problems, seizures, lack of coordination, muscle weakness

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
