# Peer review of "Antiviral Agents as Therapeutic Strategies Against Cytomegalovirus Infections"

_viruses, 2019, doi:10.3390/v12010021_

Round 1

Reviewer 1 Report

Review of "Antiviral Agents as Therapeutic Strategies against Cytomegalovirus infections" by Shiu-Jau Chen et al.

Authors reported an overview on currently available drugs for antiviral therapy against CMV infections and new potential therapeutic strategies.

The paper is performed well, with an accurate analysis of available therapeutic strategies against Cytomegalovirus infections.

Minor remarks:

Authors should explain all the abbreviations used in the text.

Antiviral drugs should be written all in the same way, some are capitalized others not...

Author Response

Reviewer 1:

Review of "Antiviral Agents as Therapeutic Strategies against Cytomegalovirus infections" by Shiu-Jau Chen et al.

Authors reported an overview on currently available drugs for antiviral therapy against CMV infections and new potential therapeutic strategies.

The paper is performed well, with an accurate analysis of available therapeutic strategies against Cytomegalovirus infections.

Minor remarks:

Authors should explain all the abbreviations used in the text.

Ans: We have provided full names of all abbreviation when they appear in the text for the first time. Additionally, we have added a list of abbreviation after the conclusions section. (P.9-10)

Antiviral drugs should be written all in the same way, some are capitalized others not.

Ans: We have checked all over the manuscript and had the names of all antiviral drugs written in the lowercase way unless they are in the beginning of a sentence.

Reviewer 2 Report

In this manuscript Chen and colleagues have reviewed the current therapeutic strategies against cytomegalovirus. This is a quite comprehensive and well-written review targeting a broad audience interested in the topic. However, I would advise the authors to delve deeper the molecular biology of cytomegaloviruses (section 1.2), giving details on viral structure, and structural proteins, which are interesting for developing vaccines, as the authors highlight in section 5, and on non-structural proteins, the main targets of antiviral drugs explained in section 3.  I believe that readers in general will appreciate a more generous introduction to these aspects because it will allow them to understand in greater depth the antiviral mechanisms described in the review. Otherwise, it's a good review that reads nicely.

Author Response

Reviewer 2:

In this manuscript Chen and colleagues have reviewed the current therapeutic strategies against cytomegalovirus. This is a quite comprehensive and well-written review targeting a broad audience interested in the topic. However, I would advise the authors to delve deeper the molecular biology of cytomegaloviruses (section 1.2), giving details on viral structure, and structural proteins, which are interesting for developing vaccines, as the authors highlight in section 5, and on non-structural proteins, the main targets of antiviral drugs explained in section 3. I believe that readers in general will appreciate a more generous introduction to these aspects because it will allow them to understand in greater depth the antiviral mechanisms described in the review. Otherwise, it's a good review that reads nicely.

Ans: We have added a new section “1.2. CV molecular biology” to provide deeper molecular biology of cytomegaloviruses including 2 new references (10, 11) and the original section 1.2 has been changed to section 1.3. (P.2-3)